# Emotional Labor, Burnout, Medical Error, and Turnover Intention among South Korean Nursing Staff in a University Hospital Setting

**DOI:** 10.3390/ijerph181910111

**Published:** 2021-09-26

**Authors:** Chan-Young Kwon, Boram Lee, O-Jin Kwon, Myo-Sung Kim, Kyo-Lin Sim, Yung-Hyun Choi

**Affiliations:** 1Department of Oriental Neuropsychiatry, Dongeui University College of Korean Medicine, Busan 47227, Korea; 2Clinical Research Coordinating Team, Korea Institute of Oriental Medicine, Daejeon 34054, Korea; qhfka9357@naver.com; 3KM Science Research Division, Korea Institute of Oriental Medicine, Daejeon 34054, Korea; cheda1334@kiom.re.kr; 4Department of Nursing, Dongeui University College of Nursing, Healthcare Sciences & Human Ecology, Busan 47340, Korea; myosg@deu.ac.kr; 5Department of Music, Pyeongtaek University Graduate School, Pyeongtaek-si 17869, Korea; shimkl@naver.com; 6Department of Biochemistry, Dongeui University College of Korean Medicine, Busan 47227, Korea; choiyh@deu.ac.kr

**Keywords:** nurse, mental health, emotional labor, burnout, medical error, turnover intention, South Korea

## Abstract

Nurses are vulnerable to mental health challenges, including burnout, as they are exposed to adverse job conditions such as high workload. The mental health of this population can relate not only to individual well-being but also to patient safety outcomes. Therefore, there is a need for a mental health improvement strategy that targets this population. This cross-sectional survey study investigates emotional labor, burnout, turnover intention, and medical error levels among 117 nursing staff members in a South Korean university hospital; it also analyzes correlations among outcomes and conduct correlation analysis and multiple regression analysis to determine relationships among these factors. The participants had moderate to high levels of emotional labor and burnout, and 23% had experienced medical errors within the last six months. Save for medical errors, all outcomes significantly and positively correlated with each other. These results can be used to improve the mental health outcomes of nurses working in the hospital and their consequences. Specifically, the job positions of nursing personnel may be a major consideration in such a strategy, and job-focused emotional labor and employee-focused emotional labor may be promising targets in ameliorating turnover intention and client-related burnout, respectively.

## 1. Introduction

Nurses are healthcare professionals who are vulnerable to various mental health challenges, including burnout, as they are exposed to adverse job characteristics such as high workload, relatively low staffing levels, and long shifts [1]. Since the mental health of healthcare professionals, including nurses, can relate not only to well-being at the individual level but also to patient safety outcomes (i.e., through medical errors and the like), there is a need to focus on the mental health of this population [2]. In addition, unfavorable mental health conditions among nurses can relate to high turnover intention; in fact, South Korean nurses are being constantly challenged with high turnover rates, and there is a need to improve what are poor working environments [3]. Additionally, although costs vary depending on the environment, nurse turnover costs pose a major concern for healthcare organizations [4]. Accordingly, the Korean Nurses Association held a nursing policy declaration ceremony in 2016 with the slogan ‘Happy Nurses Make Happy People. Securing skilled nurses for patient safety and preventing turnover was one of the main objectives. Thus, it is evident that the mental health of nurses and prevention of the resultant turnover intention are of much significance in Korea.

In developing a strategy by which to reduce turnover intention, medical error, and the potential costs thereof, emotional labor and nurse burnout may be considered promising targets. Emotional labor is exerted in managing emotions such that they align with organizational or professional display rules [5], while burnout is a state of emotional, mental, and physical exhaustion caused by excessive and prolonged stress [6]. Recently, emotional labor has been attracting attention as a factor that relates to burnout and turnover intention in nurses, and some studies report that emotional labor factors such as emotional dissonance and emotional suppression may relate to burnout [7,8,9]. High burnout in nurses is associated with high turnover intention, and empirical research is needed to better understand the so-called sustainability at work and to prevent burnout and turnover in this population [10]. Van der Heijden et al. (2019) presented a model of the order of job demands (including emotional demands), burnout, and turnover intention and emphasized the need to understand the turnover intention of nurses from the individual, organizational, and social perspectives [10]. However, few studies examine the relationships among emotional labor, burnout, turnover intention, and medical error among nurses, especially in the South Korean context. In particular, existing studies have often investigated the relationship between emotional labor, burnout, and turnover intention, but studies considering medical errors are lacking. Examinations of these factors can ultimately help relieve burdens at the individual, hospital, and societal levels by improving the individual mental health of nurses and nursing practice environments [10]. Specifically, a previous study suggested that reducing the quantitative job demands and increasing social support for nurses was associated with a decrease in turnover intention [10], suggesting the possibility of managing nurse turnover intention at the social rather than the individual level. These strategies should be developed to reflect the sociocultural context [11]. Moreover, the COVID-19 pandemic may have discriminatory effects depending on the country, suggesting that the mental health of nurses should be considered contextually [12].

The research team consists of the authors of this paper participating in a project that looks to develop strategies (i.e., smartphone application using mind-body medicine) to improve the mental health of South Korean nurses working in hospital settings. Recent findings of this project included the effectiveness of mind-body modalities on the mental health of nurses [13]. In this study, with reference to the model proposed by Van der Heijden et al. [10], the relationships between emotional labor, burnout, and turnover intention were investigated, and another important outcome, medical error, among nursing staff in a university hospital in South Korea, was also considered. This hospital is the only university hospital that provides both Western and Korean medicine services (i.e., integrative medicine), where staff are likely to be familiar with mind-body medicine in Busan, the second largest city in South Korea.

## 2. Materials and Methods

### 2.1. Participants

A cross-sectional survey was conducted among nursing staff at a university-based general hospital located in Busan. The hospital has approximately 560 beds and provides integrative medicine services in the outpatient and inpatient departments. The number of target participants was calculated using G*Power 3.1.9.2 (Heinrich-Heine-Universität Düsseldorf, Düsseldorf, Germany); for correlation analysis, the required sample size was 134 when the significance level was set to 0.05, the power to 0.95, and the effect size to 0.3. In this survey, the number of target participants was set to 150, and it was assumed that approximately 10% of the responses were of poor quality.

### 2.2. Measures

#### 2.2.1. Demographic Characteristics

The demographic characteristics investigated in this survey included gender, age, clinical experience, education level, marriage, religion, monthly income, job position, work type, work department, whether they were working in a department they wished to work, whether they were satisfied with their current department, turnover experience, and department transfer experience; subjective health status on a five-point Likert scale, anchored by *very bad* and *very good,* was also captured. It was also investigated whether working in specific shifts affected the results. The nurses here work in three shifts: day, evening, and night, starting at 7 a.m., 3 p.m., and 11 p.m., respectively.

#### 2.2.2. Emotional Labor

The emotional labor assessment tool of Lee was used to assess the study participants’ emotional labor [14]. This tool’s 14-item scale comprises two domains: employee-focused emotional labor (six items) and job-focused emotional labor (eight items). Each item is measured on a five-point Likert scale anchored by 1 (*not at all*) and 5 (*very much*); therefore, the score ranges of employee-focused emotional labor and job-focused emotional labor are 6–30 and 8–40, respectively. The higher the score was, the more severe the emotional labor. The validity of this tool was confirmed in Lee’s study in 2007 [14]. Moreover, its reliability was confirmed by Lee (not medical staff) [14] and Eo et al. (emergency room nurses) [15]. The Cronbach’s alpha vale of this tool was 0.79 in this study.

#### 2.2.3. Burnout

Ham’s Korean-language version of the Copenhagen Burnout Inventory [16] was used to assess participant burnout [17]. The 19-item scale comprises three domains: personal burnout (seven items), work-related burnout (six items), and client-related burnout (six items). Each item is measured on a five-point Likert scale anchored by 1 (*not at all*) and 5 (*very often*); therefore, the score ranges of personal burnout, work-related burnout, and client-related burnout are 7–35, 6–30, and 6–30, respectively. The higher the score was, the more severe the burnout. The Copenhagen Burnout Inventory is a useful measure to evaluate burnout, and Ham [17] and Montgomery et al. [18] confirmed its validity and reliability in nurses. The Cronbach’s alpha value of this tool was 0.94 in this study.

#### 2.2.4. Turnover Intention

The Korean Nurse Turnover Intention Scale (K-NTIS) of Yeun et al. was used to assess participant turnover intention [19]. The 10-item scale comprises three domains: job satisfaction (four items), job performance (three items), and interpersonal relationships (three items). Each item is measured on a five-point Likert scale anchored by 1 (*not at all*) and 5 (*very much*); therefore, the score ranges of job satisfaction, job performance, and interpersonal relationships are 4–20, 3–15, and 3–15, respectively. The higher the score was, the more severe the turnover intention. The validity and reliability of this tool was confirmed by Yeun et al. in Korean nurses working in general hospitals [19]. The Cronbach’s alpha vale of this tool was 0.87 in this study.

#### 2.2.5. Medical Error

To evaluate medical error, respondents were asked to respond with “yes” or “no” to whether they had experienced medical error in the six previous months. In addition, if a respondent answered “yes”, they were asked to provide further information: procedure and treatment errors, medication errors, transfusion errors, patient falls, acupuncture-related errors in KM service, and others.

### 2.3. Data Collection

This survey was conducted through an online anonymous survey (SurveyMonkey, SurveyMonkey Inc., San Mateo, CA, USA) for 10 days, from 29 April 2021 to 8 May 2021, and respondents were allowed access to the related QR code through the social networking service, and printing paper was provided to nursing staff in the hospital.

### 2.4. Data Analysis

The data collected in this study were analyzed using SAS^®^ version 9.4 (SAS Institute. Inc., Cary, NC, USA). The demographic characteristics are expressed herein as either frequencies and percentages or means and standard deviations (SDs). The levels of emotional labor, burnout, turnover intention, and medical errors were analyzed using descriptive statistics. The differences in emotional labor, burnout, turnover intention, and medical error according to demographic characteristics were further analyzed using independent *t*-tests and analysis of variance (ANOVA). To examine correlations among emotional labor, burnout, turnover intention, and medical error, Pearson’s correlation analysis was performed. The variable used for correlation analysis was the average of the scores of the Likert type questionnaire, and after confirming normality through the Shapiro–Wilk Kolmogorov–Smirnov tests, Pearson’s correlation coefficient was obtained. In addition, it was confirmed that the analysis results using the Spearman and Pearson correlation coefficients were not significantly different in this study. Finally, multiple regression analysis was performed to determine the factors that affect participant turnover intention and medical error, and checks for multicollinearity in the multiple regression analysis were carried out. Control variables were identified prior to logistic regression analysis, by using independent *t*-test or ANOVA, and Bonferroni’s post-hoc test was performed. The significance level of the statistical test was set at *p* < 0.05.

### 2.5. Ethical Considerations

The current study was conducted according to the guidelines of the Declaration of Helsinki, and the study protocols were approved by the Institutional Review Board (IRB) of Dongeui University Korean Medicine Hospital (IRB No. DH-2021-02; approved on 7 April 2021). At the beginning of the online survey, informed consent forms—which contained the details and purpose of the study and an anonymity guarantee and advised participants of their right to withdraw from participation at any time—were displayed. Only participants who consented though this informed consent form participated in this survey. All participants were given a coffee e-gift worth about KRW 4000 (approximately USD 3.50). Finally, all tools used in this study were approved for use by the respective copyright holder.

## 3. Results

### 3.1. Participants’ Demographic Characteristics

A total of 117 responses were collected and analyzed after excluding five incomplete responses (completion rate: 95.9%; 117/122). Most participants were women (*n* = 113; 96.6%) and under 30 years of age (*n* = 65; 55.6%). Most of the participants had a bachelor’s degree or higher (*n* = 87; 74.4%), and more than two-thirds (*n* = 78; 66.7%) were unmarried. Most of the participants were nonreligious (*n* = 72; 61.5%). More than one-half of the participants (*n* = 67; 57.3%) had a monthly income of less than two million KRW. Their average clinical experience (±SD) was 94.83 ± 93.25 months (range: 1–383). The most common job position among the participants was staff nurse (*n* = 63; 53.8%). In terms of work type, the ratios of shift work (*n* = 65; 55.6%) and non-shift work (*n* = 52; 44.4%) were similar. The work departments of the participants included outpatient departments (*n* = 27; 23.1%), nursing care integrated service wards (*n* = 21; 17.9%), internal medicine wards (*n* = 20; 17.1%), surgical wards (*n* = 16; 13.7%), KM departments (*n* = 12; 10.3%), operating rooms (*n* = 9; 7.7%), intensive care units (*n* = 5; 4.3%), emergency rooms (*n* = 4; 3.4%), and others (*n* = 3; 2.6%). About two-thirds of the participants (*n* = 78; 66.7%) answered that their current work department was the department where they desired to work, and most (*n* = 82; 70.1%) were satisfied with their work in the department. Most participants (*n* = 97; 82.9%) had no turnover experience. Twenty (17.1%) participants had had turnover experiences, and the average number of turnovers among them was 2.10 ± 1.80 (range: 1–9). In addition, 49 participants (41.9%) had department transfer experience, and the average number of transfers among them was 2.35 ± 2.11 (range: 1–10). Although none of the participants rated their health as very bad in their subjective health status, 27 (23.1%) rated their health as bad, 67 (57.3%) as normal, 20 (17.1%) as good, and three (2.6%) as very good (Table 1).

### 3.2. Measurement Results of the Study Participants

#### 3.2.1. Levels of Emotional Labor and Burnout

The participants’ levels of employee-focused emotional labor and job-focused emotional labor were 3.48 ± 0.54 and 3.50 ± 0.51, respectively. Overall, their level of emotional labor was 3.49 ± 0.45, suggesting a moderate to high level of emotional labor. Their levels of personal burnout, work-related burnout, and client-related burnout were 3.26 ± 0.71 (95% CI, 3.13–3.39), 3.14 ± 0.87 (95% CI, 2.98–3.30), and 2.90 ± 0.83 (95% CI, 2.75–3.05), respectively. Overall, the burnout level among the participants was 3.11 ± 0.73 (95% CI, 2.98–3.24), suggesting a moderate to high level of burnout (Appendix A).

Overall, there were statistically significant differences in emotional labor and burnout levels in terms of some sociodemographic characteristics, including education level, job position, work department, department where they wished to work, whether they were satisfied with their work, department transfer experience, and subjective health status. That is, compared to having an associate degree and lower, having a bachelor’s degree or higher related to statistically significantly higher levels of employee-focused emotional labor (*p* < 0.05), personal burnout (*p* < 0.01), work-related burnout (*p* < 0.01), and client-related burnout (*p* < 0.01). In terms of job positions, compared to being an assistant nurse (AN) or staff nurse, being a charge nurse or higher related to the highest levels of job-focused emotional labor (*p* < 0.01); however, compared to being an AN or charge nurse or higher, being a staff nurse related to the highest levels of work-related burnout (*p* < 0.05) and client-related burnout (*p* < 0.01). Regarding work departments, the level of job-focused emotional labor was highest among participants included in other categories, such as administrative positions (*p* < 0.05). Working in a work department in which one did not wish to work related to statistically significantly higher levels of employee-focused emotional labor (*p* < 0.05), job-focused emotional labor (*p* < 0.05), personal burnout (*p* < 0.05), and work-related burnout (*p* < 0.01). Working in a work department in which one was not satisfied was also related to statistically significantly higher levels of personal burnout (*p* < 0.01), work-related burnout (*p* < 0.01), and client-related burnout (*p* < 0.01). Having a department transfer experience was related to a statistically significantly higher level of job-focused emotional labor (*p* < 0.01). Finally, compared to normal or good subjective health status, poor subjective health status related to the highest level of employee-focused emotional labor (*p* < 0.01), personal burnout (*p* < 0.01), work-related burnout (*p* < 0.01), and client-related burnout (*p* < 0.01).

#### 3.2.2. Medical Error and Turnover Intention Levels

Twenty-seven (23.1%) of the participants answered that they had experienced 40 medical errors within the six previous months, including procedure and treatment errors (*n* = 5; 18.5%), medication errors (*n* = 14; 51.9%), transfusion errors (*n* = 3; 11.1%), patient falls (*n* = 11; 40.7%), acupuncture-related errors in KM service (*n* = 3; 11.1%), and others (*n* = 4; 14.8%). The job satisfaction, job performance, and interpersonal relationship levels in terms of turnover intention were 3.91 ± 0.78, 3.79 ± 0.74, and 3.69 ± 0.75, respectively. Overall, the participants’ turnover intention level was 3.81 ± 0.65, suggesting a moderate to high turnover intention level (Appendix A).

There were statistically significant differences in levels of medical error and turnover intention according to some sociodemographic characteristics (i.e., education level, monthly income, clinical experience, job position, being in a department where they were satisfied with work, turnover experience, and subjective health status). That is, compared to having an associate degree or lower, having a bachelor’s degree or higher related to statistically significantly higher levels of turnover intention (*p* < 0.01). Compared to having a monthly income of less than 2.5 million won, having a monthly income of 2.5 million won or more related to a statistically significantly higher turnover intention (*p* < 0.01). Compared to having six or more years of clinical experience, having less than six years of clinical experience significantly related to making frequent medical errors (*p* < 0.05). Compared to being an AN or a charge nurse or higher, being a staff nurse related to making medical errors more frequently (*p* < 0.05); however, compared to being an AN or staff nurse, being a charge nurse or higher related to the highest levels of turnover intention (*p* < 0.01). Both working in departments where one was not satisfied with work (*p* < 0.01) and having turnover experience (*p* < 0.05) related to significantly higher levels of turnover intention. Finally, compared to having a normal or good subjective health status, having a bad subjective health status related to making medical errors more frequently (*p* < 0.05) (Appendix A).

However, there were no statistically significant differences in all outcomes, including emotional labor, burnout, medical error, and turnover intention levels in terms of age, marital status, religion, and type of work (shift or non-shift) (all *p* > 0.05) (Appendix A).

### 3.3. Correlations among the Outcomes

Except for a significant and positive correlation between work-related burnout and medical error (r = 0.20, *p* < 0.05), no outcome showed a significant association with medical error. On the other hand, the remaining outcomes showed a significant and positive correlation with each other in all respects. There were significant and positive correlations between subscales of emotional labor (i.e., employee-focused emotional labor and job-focused emotional labor) and subscales of burnout (i.e., personal, work-related, and client-related burnout) (r = 0.24 to 0.90, *p* < 0.05 or < 0.01). There were also significant and positive correlations between levels of emotional labor, burnout, and turnover intention (employee-focused emotional labor: r = 0.23, *p* < 0.05; job-focused emotional labor: r = 0.33, *p* < 0.01; personal burnout: r = 0.28, *p* < 0.01; work-related burnout: r = 0.31, *p* < 0.01; client-related burnout: r = 0.27, *p* < 0.01) (Table 2).

### 3.4. Effects of Emotional Labor and Burnout on Turnover Intention and Medical Error

In the multiple regression analysis, all regression conditions were verified before running the analysis. Because most of the standardized errors have values between −2 and 2, it is confirmed that there are no outliers that deviate significantly from the model estimates. In addition, as a result of confirming the equality of variance using the chi-square test value for the variance-covariance matrix, all models satisfied the equality of variance. Since the variance inflation factor of all variables except the dummy variables was less than 10, it was confirmed that there was no multicollinearity. The Durbin–Watson statistic of all models has a minimum value of 1.70 and a maximum of 2.11, so independence is satisfied. As a result of examining the Shapiro–Wilk and Kolmogorov–Smirnov tests for error, normality was also satisfied. Finally, it was confirmed that there were no influential cases because there were no cases where Cook’s distance was greater than 1 in all observations.

To evaluate the effects of emotional labor and burnout on turnover intention and medical error, a basic model was constructed based on the demographic characteristics that were found to have significant effects on emotional labor, burnout turnover, intention, and/or medical error. Employee-focused emotional labor (Model 1), job-focused emotional labor (Model 2), personal burnout (Model 3), work-related burnout (Model 4), and client-related burnout (Model 5) were added to the basic model, and multiple regression analysis was performed to evaluate their significance. According to the findings, in Model 1, employee-focused emotional labor significantly positively impacted total burnout, with an explanatory power of 35.8% (β = 0.28, *p* < 0.05), and client-related burnout positively, with an explanatory power of 27.7% (β = 0.42, *p* < 0.01). Moreover, in Model 2, job-focused emotional labor had a significant positive impact on turnover intention, with an explanatory power of 29.2% (β = 0.28, *p* < 0.05). Otherwise, neither emotional labor nor burnout were found to have any significant effect on turnover intention or medical error (Table 3 and Table 4).

## 4. Discussion

### 4.1. Clinical Interpretations

The results of this survey indicated that the participants had moderate to high levels of emotional labor, burnout, and turnover intention. Approximately one in five nursing staff had experienced a medical error within the six previous months, and so medical errors were not uncommon. According to the correlation analysis results, all outcomes except for medical error showed a significant positive correlation with each other—that is, adverse mental health outcomes and higher turnover intentions are likely to coexist. These findings align with those of previous studies [20,21,22]. Interestingly, being a staff nurse was associated with higher levels of work-related burnout and client-related burnout than being a charge nurse, and being a charge nurse was associated with higher levels of job-focused emotional labor. In addition, being a staff nurse was associated with more frequent medical errors, and being a charge nurse was associated with higher turnover intention. Because staff nurses provide nursing services while interacting with a variety of patients on the front line, being one can be associated with a high level of burnout and medical error. Charge nurses, on the other hand, must make relatively more important decisions while interacting with “fussier” or more important patients; therefore, they may be associated with higher levels of emotional labor and turnover intention. This suggests that the various job positions among nursing staff should be considered when establishing strategies by which to improve their otherwise adverse mental health and mitigate its consequences.

Among the study participants, both being in a work department where one did not desire to work and being in a work department where one did not derive satisfaction correlated with higher levels of emotional labor, burnout, and/or turnover intention, but not medical error. These two factors can be improved at the hospital level, and so it is possible to use strategies to regularly assess work desire and satisfaction and incorporate them into their work, with the endpoints of improving otherwise adverse mental health and the consequences thereof. In addition, a bad subjective health status correlated with higher levels of emotional labor and burnout and with medical error experience, suggesting that the provision of health promotion programs targeting nursing staff within a university hospital stands as a promising strategy by which to improve nurses’ mental health and patient safety. According to a recent systematic review [23], lifestyle health promotion interventions that target nurses can potentially have a positive effect on their health and well-being. In light of these findings, it is asserted that it is possible to extend the impact of lifestyle health promotion interventions.

According to the correlation analysis results, there is a significant positive correlation between work-related burnout and medical error; however, according to the regression analysis results, work-related burnout does not significantly affect the occurrence of medical error. In other words, the findings suggests that improvement strategies in each of these two are needed independently. However, in the case of turnover intention, job-focused emotional labor may be a promising target in a strategy by which to reduce turnover intention among nursing staff since job-focused emotional labor was found in the regression analysis to have a significant effect. For the same reason, employee-focused emotional labor may also be a promising target in improving client-related burnout. According to a recent meta-analysis of predictors of hospital nurse turnover intention in South Korea, burnout, emotional exhaustion, job stress, and career plateau showed positive effects, while organizational commitment, individual–organization fit, career commitment, work engagement, job satisfaction, and job embeddedness showed negative effects [24]. The findings suggest that among the effects on turnover intention, job-focused emotional labor may be most important. In addition, the relationship between emotional labor and burnout, as found in a previous study [20], was further refined as the relationship between employee-focused emotional labor and client-related burnout.

### 4.2. Results of Previous Studies

Previous studies have investigated the emotional labor, burnout, and turnover intention of Korean nurses. For example, Back et al. [20] found a mediating role of burnout in the relationship between some aspects of emotional labor and turnover intention in clinical nurses in Korea. Interestingly, in their study, emotional demand and regulation, a type of emotional labor, did not have a direct relationship with turnover intention [20]. In other words, it is suggested that the effect on turnover intention may differ depending on the type of emotional labor. It is noted that although there is currently insufficient empirical evidence to support these results, this may be because nurses may take emotional demand and regulation for granted in their job [20]. In the current study, job-focused emotional labor significantly impacted turnover intention, while employee-focused emotional labor did not. Employee-focused emotional labor consists of superficial and deep acting, in which the former is the act of expressing emotions that are not actually felt, and the latter is actually experiencing the emotions that one is supposed to express [14], which share similar aspects with emotional demand and regulation. On the other hand, job-focused emotional labor evaluates the job environment that creates the pressure to perform emotional labor [14]. Therefore, the result of the current study that only job-focused emotional labor had a significant impact on turnover intention suggests the possibility of reducing turnover intention of nursing staff by improving the job environment at the hospital level. Although the findings provide empirical data that partially support the results of Back et al. [20], the impacts of superficial and deep acting on turnover intention in nurses need to be further investigated with a larger sample size.

One of the unique aspects of this study is that it used medical error, along with emotional labor, burnout, and turnover intention, which has rarely been considered together in previous Korean studies. Recent studies have provided useful insights in this context. Park et al. [25] investigated the relationship between burnout, job satisfaction, turnover intentions, and medical errors among nurses in integrated nursing care wards. Interestingly, regression analysis in their study found that medical error was significantly associated with nursing care left undone (i.e., omission or failure to complete all or part of nursing care to be performed within working hours due to lack of time) and nursing experience, but not burnout, job satisfaction, or turnover intention [25]. In our current regression analysis, emotional labor and burnout did not have a significant impact on medical error, suggesting that an organizational strategy such as reducing nursing care left undone may be needed to potentially reduce medical error, i.e., medical error might be affected by factors existing in environmental contexts other than emotional labor and burnout, and additional research is needed to explore this.

Contrary to what might be expected, no significant differences were found in emotional labor, burnout, medical error, and turnover intention levels according to the type of work (i.e., shift work or non-shift work). However, Lee et al. [26] suggested that working night shifts is a risk factor for burnout in nurses working in general hospitals. Another study found that shift nurses experienced sickness presenteeism more frequently than non-shift nurses [27]. One possibility for this discrepancy is that shift work may be associated with poor mental health through the mediation of sleep disturbances that were not investigated in this study [28]. Other possibilities include the potential mediation of other contributors to shift work tolerance, including age, morningness, self-esteem, social support, job stress, alcohol consumption, physical activity, and number of night shifts [29].

### 4.3. Study Limitations

The following limitations should be considered when interpreting our results. First, this study was conducted at a university hospital in South Korea. This hospital is unique in that it is the only university hospital that provides integrative medicine services in Busan, the second largest city in South Korea. However, this also means that these participants do not represent the entire Korean nursing staff population. In addition, the sample size calculated before the survey was set to 150, in consideration of the potential poor response, but the final number of participants was 117, which is less than the required sample size of 134. The researchers requested cooperation from the nursing headquarters of this hospital on the 5th day of the survey to encourage participation of nursing staff, but the participation rate did not increase significantly. Therefore, through in-depth discussion among the researchers, the survey ended on the 10th day because it was thought that ongoing pressure to participate in the study could aggravate the nurses’ work stress. Therefore, it is acknowledged as a major limitation that this research did not reach the target sample size. Second, this study was conducted as a cross-sectional survey. In the spring and summer of 2021, when this survey was conducted, South Korea saw the severe effects of the COVID-19 pandemic. Since this pandemic could have had a significant impact on the mental health of nurses [30], the possibility cannot be ruled out that the participants’ emotional labor, burnout, medical errors, and turnover intentions were affected by the COVID-19 pandemic. Third, the sample in this survey was not sufficiently large to make its results generalizable; having such a small sample also leads to insufficient subgroup analysis. In particular, although previous studies show that a nurse’s work department is a factor that can have a significant impact on their mental health, according to the regression analysis results, the work department had no significant effect on any outcome. However, among the study participants, fewer than 10 nurses worked in an intensive care unit, operating room, or emergency room (3.4–7.7%). This divergence from the results of previous studies may stem from the small sample size of this study. Fourth, a recent meta-analysis synthesized predictors of turnover intention in Korean hospital nurses and reported that emotional exhaustion, depression, job stress, fatigue, and emotional labor were the facilitating factors, while work engagement, job satisfaction, and professional self-concept were the preventative factors [24]. This suggests that the turnover intention of this population may be influenced by various currently existing personal characteristics and occupational perceptions that were not considered in this study. For example, depression is an important mental health variable at the individual level, but the current survey did not collect data to adjust for individual depression. Therefore, the possibility that the results of this study may have been influenced by potentially unexamined variables cannot be excluded.

## 5. Conclusions

In conclusion, the data captured through the cross-sectional survey suggest that nursing staff in a Busan-based university hospital had moderate to high levels of emotional labor, burnout, and turnover intention. Approximately one in five nursing staff members there had experienced a medical error within the six previous months. Job position may be a major consideration in mental health improvement strategies that target nursing staff; additionally, job-focused emotional labor (by which to improve turnover intention) and employee-focused emotional labor (by which to improve client-related burnout) may be promising targets. Nonetheless, the ability to generalize the findings is weak, given that the hospital-setting sample size is small; for this reason, future research should undertake a multicenter study with a larger sample size.

## Figures and Tables

**Table 1 ijerph-18-10111-t001:** Demographic characteristics of participants (*N* = 117).

Features	Category	*N*	%
Gender	Male	4	3.4
	Female	113	96.6
Age (years)	<30	65	55.6
	≥30	52	44.4
Education level	Associate degree or below	30	25.6
	Bachelor degree or above	87	74.4
Marriage	Unmarried	78	66.7
	Married	37	31.6
	Other (divorced, separated, widowed)	2	1.7
Religion	Christianity (Protestant)	15	12.8
	Buddhism	26	22.2
	Catholicism	1	0.9
	No religion	72	61.5
	Other	3	2.6
Monthly income	<2.5 million KRW	67	57.3
	≥2.5 million KRW	50	42.7
Clinical experience (years)	<6	70	59.8
	≥6	47	40.2
	Mean ± SD (month)	94.83 ± 93.25	Range: 1–383
Job position	Assistant nurse	18	15.4
	Staff nurse	63	53.8
	Charge nurse or higher	36	30.8
Type of work	Shift	65	55.6
	No shift	52	44.4
Work department	WM OPD and IPD (IM, surgical, OR)	72	61.5
	ER and ICU	9	7.7
	NCISW	21	17.9
	KM OPD and IPD	12	10.3
	Other	3	2.6
In the department where they wished to work	No	39	33.3
	Yes	78	66.7
In a department where they are satisfied with the work	No	35	29.9
	Yes	82	70.1
Turnover experience	No	97	82.9
	Yes	20	17.1
	Mean ± SD (*n*)	2.10 ± 1.80	Range: 1–9
Department transfer experience	No	68	58.1
	Yes	49	41.9
	Mean ± SD (*n*)	2.35 ± 2.11	Range: 1–10
Subjective health status	Very bad	0	0
	Bad	27	23.1
	Normal	67	57.3
	Good	20	17.1
	Very good	3	2.6

Abbreviations. ER, emergency room; ICU, intensive care unit; IM, internal medicine; IPD, inpatient department; KM, Korean medicine; NCISW, nursing care integrated service ward; OPD, outpatient department; OR, operating room; SD, standard deviation; WM, Western medicine.

**Table 2 ijerph-18-10111-t002:** Correlations between emotional labor, burnout, turnover intention, and medical error (*N* = 117).

Variable	Pearson Correlation (*p*-Value)						
	1	2	3	4	5	6	7
1. Employee-focused emotional labor	1	0.442 (<0.0001)	0.341 (0.0002)	0.334 (0.0002)	0.378 (<0.0001)	0.104 (0.2627)	0.225 (0.0148)
2. Job-focused emotional labor		1	0.276 (0.0026)	0.237 (0.0100)	0.246 (0.0075)	0.148 (0.1112)	0.334 (0.0002)
3. Personal burnout			1	0.890 (<0.0001)	0.637 (<0.0001)	0.178 (0.0548)	0.279 (0.0023)
4. Work-related burnout				1	0.761 (<0.0001)	0.196 (0.0341)	0.307 (0.0008)
5. Client-related burnout					1	0.120 (0.1969)	0.270 (0.0033)
6. Medical error						1	0.051 (0.5823)
7. Turnover intention							1

**Table 3 ijerph-18-10111-t003:** Influence of emotional labor on burnout, medical error, and turnover intention (N = 117).

	Burnout Parameter Estimates (*p*-Value)				Medical Error Parameter Estimates (*p*-Value)	Turnover Intention Parameter Estimates (*p*-Value)
	Personal	Work-Related	Client-Related	Total		
F-value (*p*-value)	4.08 (<0.0001)	4.76 (<0.0001)	3.78 (<0.0001)	5.04 (<0.0001)		2.33 (0.0057)
R^2^ (Adj R^2^)	0.395 (0.298)	0.433 (0.342)	0.377 (0.277)	0.446 (0.358)		0.272 (0.155)
Intercept (Constant)	2.060 (0.0016)	1.52 (0.0467)	0.648 (0.3902)	1.449 (0.0228 *)	−5.529 (0.0484)	3.200 (<0.0001)
Employee-focused emotional labor	0.213 (0.0673)	0.235 (0.0889)	0.419 (0.0027 *)	0.284 (0.0141 *)	0.731 (0.1746)	0.144 (0.2198)
F-value (*p*-value)	3.85 (<0.0001)	4.46 (<0.0001)	2.94 (0.0005)	2.58 (0.0022)		2.58 (0.0022)
R^2^ (Adj R^2^)	0.381 (0.282)	0.417 (0.323)	0.320 (0.212)	0.292 (0.179)		0.292 (0.179)
Intercept (Constant)	2.194 (0.0043)	2.024 (0.0255)	1.597 (0.0846)	1.955 (0.0108)	−7.414 (0.0227)	2.543 (0.0008 *)
Job-focused emotional labor	0.138 (0.3059)	0.063 (0.6930)	0.102 (0.5337)	0.103 (0.4448)	1.103 (0.0788)	0.277 (0.0378 *)

Note: Covariates are omitted from this table, and the full data are presented in Appendix A. Abbreviations. AN, assistant nurse; ER, emergency room; ICU, intensive care unit; IPD, inpatient department; KM, Korean medicine; NCISW, nursing care integrated service ward; OPD, outpatient department; WM, Western medicine. * *p* < 0.05.

**Table 4 ijerph-18-10111-t004:** Influence of burnout on turnover and medical error (*N* = 117).

	Medical Error Parameter Estimates (*p*-Value)				Turnover Intention Parameter Estimates (*p*-Value)			
F-value (*p*-value)					2.32 (0.0059 *)	2.29 (0.0066 *)	2.23 (0.0084 *)	2.30 (0.0066 *)
R^2^ (Adj R^2^)					0.271 (0.154)	0.268 (0.151)	0.263 (0.145)	0.269 (0.152)
Intercept (Constant)	−4.802 (0.0529)	−4.498 (0.0519)	−3.547 (0.1129)	−4.483 (0.0599)	3.342 (<0.0001 *)	3.470 (<0.0001 *)	3.571 (<0.0001 *)	3.421 (<0.0001 *)
Personal burnout	0.57 (0.1768)				0.118 (0.2369)			
Work-related burnout		0.568 (0.1280)				0.086 (0.3072)		
Client-related burnout			0.208 (0.5482)				0.048 (0.5618)	
Burnout				0.546 (0.1971)				0.104 (0.2992)

Note: Covariates are omitted from this table, and the full data are presented in Appendix A. Abbreviations. AN, assistant nurse; ER, emergency room; ICU, intensive care unit; IPD, inpatient department; KM, Korean medicine; NCISW, nursing care integrated service ward; OPD, outpatient department; WM, Western medicine. * *p* < 0.05.

## Data Availability

The datasets used or analyzed during the current study are available from the corresponding author upon reasonable request.

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
