# Peer review of "Emotional Labor, Burnout, Medical Error, and Turnover Intention among South Korean Nursing Staff in a University Hospital Setting"

_ijerph, 2021, doi:10.3390/ijerph181910111_

Round 1

Reviewer 1 Report

Thank you to the journal for the review possibility of the article and I appreciate the authors for the efforts they have made so far.

I think your work is interesting, dealing with issues related to emotional labor, burnout, medical error, and turnover intention, but is not yet to be published at this stage as article has serious flaws. Aspects number one and three are minor and the second is a major aspect to be considered:

1. Introduction: I suggest that you continue from why this study is important and explain in the introduction the structure of your paper, methods and previous debates in literature addressing the studied aspects. In this sense, it is important to present the different approaches that have been given by other researchers and that you built your research on, latter introduced. Here you may find one article that developed a model, explained the methods and connected this model to literature: https://www.mdpi.com/1660-4601/16/11/2011/htm (Van der Heijden, B.; Brown Mahoney, C.; Xu, Y. Impact of Job Demands and Resources on Nurses’ Burnout and Occupational Turnover Intention Towards an Age-Moderated Mediation Model for the Nursing Profession. Int. J. Environ. Res. Public Health 201916, 2011. https://doi.org/10.3390/ijerph16112011). Please consult this reference- a recent debate concerning your topic - understanding of so-called sustainability at work, in particular of how to prevent burnout and turnover among nurses, from the present International Journal of Environmental Research and Public Health.

2. One major concern is that it is a small study based on a small number of respondents. At the Participants section, in the lines 65-72, the authors mention that The number of target participants was calculated and the required sample size was 134 when the significance level was set to 0.05, the power to 0.95, and the effect size to 0.3. In this survey, the number of target participants was set to 150, and we assumed approximately 10% of the responses to be of poor quality.... Finally, 117 nursing staff members in only one hospital were considered. Why this only one hospital is interesting & relevant and why there were not considered more participants so to reach a relevant number, the targeted one (line 65-72). For the correlations among outcomes, you used Pearson (P) but did not justify why, why not Spearman (S) , normal data- Pearson, in case of non-normal data- Spearman..Regarding the multiple regression analysis, it is not confirmed that all regression conditions were verified before running the analysis, only multicollinearity is mentioned, but the following are not: singularity, homoscedasticity, outliers avoidance, Tolerance values, normality, influential cases, additivity and linearity. See Field, A. (2013). Discovering Statistics Using IBM SPSS Statistics. New Delhi: SAGE Publiations Inc.

3.More relevant literature in the field should be discussed as in the present form it does not yet engage sufficiently with recent debates in the theoretical literature, including this journal debates. More should be built on an appropriate base of theory and concepts, what are the debates, previous research on the studied constructs.

Due to these reasons, please take the feedback as a constructive help on your journey to turning this research into a good paper.

Hope my comments are helpful to improve the quality of the manuscript. Good luck!

Author Response

  • Response to Comments from Reviewer 1

Overall comment:

Thank you to the journal for the review possibility of the article and I appreciate the authors for the efforts they have made so far.

I think your work is interesting, dealing with issues related to emotional labor, burnout, medical error, and turnover intention, but is not yet to be published at this stage as article has serious flaws. Aspects number one and three are minor and the second is a major aspect to be considered:  

Response:              

Thank you for your thorough review and informative comments. We did our best to incorporate your comments into the manuscript in this revision, and we welcome your additional comments.

Comment 1:

  1. Introduction: I suggest that you continue from why this study is important and explain in the introduction the structure of your paper, methods and previous debates in literature addressing the studied aspects. In this sense, it is important to present the different approaches that have been given by other researchers and that you built your research on, latter introduced. Here you may find oone article that developed a model, explained the methods and connected this model to literature: https://www.mdpi.com/1660-4601/16/11/2011/htm (Van der Heijden, B.; Brown Mahoney, C.; Xu, Y. Impact of Job Demands and Resources on Nurses’ Burnout and Occupational Turnover Intention Towards an Age-Moderated Mediation Model for the Nursing Profession. Int. J. Environ. Res. Public Health 2019, 16, 2011. https://doi.org/10.3390/ijerph16112011). Please consult this reference- a recent debate concerning your topic - understanding of so-called sustainability at work, in particular of how to prevent burnout and turnover among nurses, from the present International Journal of Environmental Research and Public Health.

Response:              

Thank you for your comment. We have reviewed the literature you recommended, and based on these documents, we have strengthened the Introduction section. In particular, we have further emphasized the importance of this study, its rationale, and described our study's method in the context of existing approaches to this topic.

“In developing a strategy by which to reduce turnover intention, medical error, and the potential costs thereof, emotional labor and nurse burnout may be considered promising targets. Emotional labor is exerted in managing emotions such that they align with organizational or professional display rules [5], while burnout is a state of emotional, mental, and physical exhaustion caused by excessive and prolonged stress [6]. Recently, emotional labor has been attracting attention as a factor that relates to burnout and turnover intention in nurses, and some studies report that emotional labor factors such as emotional dissonance and emotional suppression may relate to burnout [7-9]. High burnout in nurses is associated with high turnover intention, and empirical research is needed to better understand the so-called sustainability at work and to prevent burnout and turnover in this population [10]. Van der Heijden et al. (2019) presented a model of the order of job demands (including emotional demands), burnout, and turnover intention, and emphasized the need to understand the turnover intention of nurses from the individual, organizational, and social perspectives [10]. However, few studies examine the relationships among emotional labor, burnout, turnover intention, and medical error among nurses, especially in the South Korean context. In particular, existing studies have often investigated the relationship between emotional labor, burnout, and turnover intention, but studies considering medical errors are lacking. Examinations of these factors can ultimately help relieve burdens at the individual, hospital, and societal levels by improving the individual mental health of nurses and nursing practice environments [10]. Specifically, a previous study suggested that reducing the quantitative job demands and increasing social support for nurses was associated with a decrease in turnover intention [10], suggesting the possibility of managing nurse turnover intention at the social rather than the individual level. These strategies should be developed to reflect the sociocultural context [11]. Moreover, the COVID-19 pandemic may have discriminatory effects depending on the country, suggesting that the mental health of nurses should be considered contextually [12].”

(please see page 2, red words)

Comment 2:

  1. One major concern is that it is a small study based on a small number of respondents. At the Participants section, in the lines 65-72, the authors mention that The number of target participants was calculated and the required sample size was 134 when the significance level was set to 0.05, the power to 0.95, and the effect size to 0.3. In this survey, the number of target participants was set to 150, and we assumed approximately 10% of the responses to be of poor quality.... Finally, 117 nursing staff members in only one hospital were considered. Why this only one hospital is interesting & relevant and why there were not considered more participants so to reach a relevant number, the targeted one (line 65-72). For the correlations among outcomes, you used Pearson (P) but did not justify why, why not Spearman (S) , normal data- Pearson, in case of non-normal data- Spearman. Regarding the multiple regression analysis, it is not confirmed that all regression conditions were verified before running the analysis, only multicollinearity is mentioned, but the following are not: singularity, homoscedasticity, outliers avoidance, Tolerance values, normality, influential cases, additivity and linearity. See Field, A. (2013). Discovering Statistics Using IBM SPSS Statistics. New Delhi: SAGE Publiations Inc.

Response:              

Thank you for the comment.

  1. As stated in the Introduction, we are in the process of developing strategies (i.e., smartphone applications) to improve mental health in this hospital. To that end, we will be able to evaluate the mental health status of nurses in this hospital and use it to develop a smartphone application tailored to this group in general.

“The research team consists of the authors of this paper participating in a project that looks to develop strategies (i.e., smartphone application using mind-body medicine) to improve the mental health of South Korean nurses working in hospital settings. Recent findings of this project included the effectiveness of mind-body modalities on the mental health of nurses [13]. In this study, with reference to the model proposed by Van der Heijden et al. [10], the relationships between emotional labor, burnout, and turnover intention were investigated, and another important outcome, medical error, among nursing staff in a university hospital in South Korea, was also considered. This hospital is the only university hospital that provides both Western and Korean medicine services (i.e., integrative medicine), where staff are likely to be familiar with mind-body medicine in Busan, the second largest city in South Korea.”

(please see page 2, red words)

  1. This is the only university hospital that provides both Western and Korean medicine (KM) services (i.e., integrative medicine) in Busan, the second largest city in South Korea. We requested cooperation from the nursing headquarters of this hospital on the 5th day of the survey to encourage participation of nursing staff, but the participation rate did not increase significantly. Therefore, after in-depth discussion among the researchers, the survey ended on the 10th day because it was thought that ongoing pressure to participate in the study could aggravate the nurses’ work stress. Therefore, it is recognized as a major limitation that this survey did not reach the target sample size. We emphasize these characteristics and limitations in the Introduction and Discussion sections.

First, this study was conducted at a university hospital in South Korea. This hospital is unique in that it is the only university hospital that provides integrative medicine services in Busan, the second largest city in South Korea. However, this also means that these participants do not represent the entire Korean nursing staff population. In addition, the sample size calculated before the survey was set to 150, in consideration of the potential poor response, but the final number of participants was 117, which is less than the required sample size of 134. The researchers requested cooperation from the nursing headquarters of this hospital on the 5th day of the survey to encourage participation of nursing staff, but the participation rate did not increase significantly. Therefore, through in-depth discussion among the researchers, the survey ended on the 10th day because it was thought that ongoing pressure to participate in the study could aggravate the nurses’ work stress. Therefore, it is acknowledged as a major limitation that this research did not reach the target sample size.”

(please see page 15, red words)

  1. The variable used in the correlation analysis was the average of the scores of the Likert scale questionnaire, and after confirming normality through the Shapiro-Wilk test and the Kolmogorov-Smirnov test, Pearson's correlation coefficient was obtained. In addition, it was confirmed that the analysis results using the Spearman and Pearson correlation coefficients were not significantly different.

“To examine correlations among emotional labor, burnout, turnover intention, and medical error, Pearson’s correlation analysis was performed. The variable used for correlation analysis was the average of the scores of the Likert type questionnaire, and after confirming normality through the Shapiro-Wilk Kolmogorov-Smirnov tests, Pearson's correlation coefficient was obtained. In addition, it was confirmed that the analysis results using the Spearman and Pearson correlation coefficients were not significantly different in this study.”

(please see page 4, red words)

  1. Because most of the standardized errors have values between -2 and 2, it is confirmed that there are no outliers that deviate significantly from the model estimates. In addition, as a result of confirming the equality of variance using the chi-square test value for the variance-covariance matrix, all models satisfied the equality of variance. Since the variance inflation factor of all variables except the dummy variables was less than 10, it was confirmed that there was no multicollinearity. The Durbin-Watson statistic of all models has a minimum value of 1.70 and a maximum of 2.11, so independence is satisfied. As a result of examining the Shapiro-Wilk and Kolmogorov-Smirnov tests for error, normality was also satisfied. Finally, it was confirmed that there were no influential cases because there were no cases where Cook's distance was greater than 1 in all observations.

“In the multiple regression analysis, all regression conditions were verified before running the analysis. Because most of the standardized errors have values between -2 and 2, it is confirmed that there are no outliers that deviate significantly from the model estimates. In addition, as a result of confirming the equality of variance using the chi-square test value for the variance-covariance matrix, all models satisfied the equality of variance. Since the variance inflation factor of all variables except the dummy variables was less than 10, it was confirmed that there was no multicollinearity. The Durbin-Watson statistic of all models has a minimum value of 1.70 and a maximum of 2.11, so independence is satisfied. As a result of examining the Shapiro-Wilk and Kolmogorov-Smirnov tests for error, normality was also satisfied. Finally, it was confirmed that there were no influential cases because there were no cases where Cook's distance was greater than 1 in all observations.”

(please see page 9, red words)

Comment 3:

3.More relevant literature in the field should be discussed as in the present form it does not yet engage sufficiently with recent debates in the theoretical literature, including this journal debates. More should be built on an appropriate base of theory and concepts, what are the debates, previous research on the studied constructs.

Response:              

Thank you for your comment. We further strengthened recent debates in the discussion section.

“4.2. Results of previous studies

Previous studies have investigated the emotional labor, burnout, and turnover intention of Korean nurses. For example, Back et al. [20] found a mediating role of burnout in the relationship between some aspects of emotional labor and turnover intention in clinical nurses in Korea. Interestingly, in their study, emotional demand and regulation, a type of emotional labor, did not have a direct relationship with turnover intention [20]. In other words, it is suggested that the effect on turnover intention may differ depending on the type of emotional labor. It is noted that although there is currently insufficient empirical evidence to support these results, this may be because nurses may take emotional demand and regulation for granted in their job [20]. In the current study, job-focused emotional labor significantly impacted turnover intention, while employee-focused emotional labor did not. Employee-focused emotional labor consists of superficial and deep acting, in which the former is the act of expressing emotions that are not actually felt, and the latter is actually experiencing the emotions that one is supposed to express [14], which share similar aspects with emotional demand and regulation. On the other hand, job-focused emotional labor evaluates the job environment that creates the pressure to perform emotional labor [14]. Therefore, the result of the current study that only job-focused emotional labor had a significant impact on turnover intention, suggests the possibility of reducing turnover intention of nursing staff by improving the job environment at the hospital level. Although the findings provide empirical data that partially support the results of Back et al. [20], the impacts of superficial and deep acting on turnover intention in nurses need to be further investigated with a larger sample size.

One of the unique aspects of this study is that it used medical error, along with emotional labor, burnout, and turnover intention, which has rarely been considered together in previous Korean studies. Recent studies have provided useful insights in this context. Park et al. [25] investigated the relationship between burnout, job satisfaction, turnover intentions, and medical errors among nurses in integrated nursing care wards. Interestingly, regression analysis in their study found that medical error was significantly associated with nursing care left undone (i.e., omission or failure to complete all or part of nursing care to be performed within working hours due to lack of time) and nursing experience, but not burnout, job satisfaction, and turnover intention [25]. In our current regression analysis, emotional labor and burnout did not have a significant impact on medical error, suggesting that an organizational strategy such as reducing nursing care left undone may be needed to potentially reduce medical error; i.e., medical error might be affected by factors existing in environmental contexts other than emotional labor and burnout, and additional research is needed to explore this.

Contrary to what might be expected, no significant differences were found in emotional labor, burnout, medical error, and turnover intention levels according to the type of work (i.e., shift work or non-shift work). However, Lee et al. [26] suggested that working night shifts is a risk factor for burnout in nurses working in general hospitals. Another study found that shift nurses experienced sickness presenteeism more frequently than non-shift nurses [27]. One possibility for this discrepancy is that shift work may be associated with poor mental health through the mediation of sleep disturbances that were not investigated in this study [28]. Other possibilities include the potential mediation of other contributors to shift work tolerance, including age, morningness, self-esteem, social support, job stress, alcohol consumption, physical activity, and number of night shifts [29].”

(please see page 14, red words)

Comment 4:

Due to these reasons, please take the feedback as a constructive help on your journey to turning this research into a good paper.

Hope my comments are helpful to improve the quality of the manuscript. Good luck!

Response:              

Thank you for your comment. I think your comments have made our research more valuable.

Reviewer 2 Report

This paper describes a part of a larger project assessing factors related to burnout, emotional labor, medical errors and turnover intention in nurses in South Korea.  The paper addresses issues that are pertinent to healthcare, and it includes new tasks that have not been evaluated in the past (as well as a novel population).

Overall the paper has some merit, but there are some major issues to be addressed. The biggest concern is lack of background and support for the measures, as well as the small sample size. In addition, one of the biggest factors related to burnout in healthcare is shift-work, and that is not mentioned at all.  For the participants, the type of certification is listed, but we know nothing about the shift types, duration, frequency, etc.  That is a major factor that should be considered.

Specific details are listed below:

  1. Lit review is too short. They mention turnover intention and the difficult job nurses have, but they suddenly jump to medical errors and burnout without talking about history related to factors affecting those.  Definitions of emotional labor and burnout, as well as previous studies tying them together and detailing how they affect nurses, should be expanded.
  2. Line 55“Examinations of these factors can ultimately help relieve burdens at the individual, hospital, and societal levels by improving the individual mental health of nurses and nursing practice environments.”  With this statement, you assume we are aware of burdens at individual, hospital and societal levels, but you have provided no information.
  3. Line 58“Our research team is participating in a project that looks to develop strategies by which to improve the mental health of South Korean nurses working in hospital settings.” This information needs to be touched on in the lit review. There is no background provided on this or what mental health issues are in South Korean nurses.
  4. Line 58 paragraph starts using first person, which is unusual in a primary research article
  5. Why doesn’t the participants section have the actual number of participants and demographics on them?
  6. The emotional labor measure should have more information on the reliability and validity of the measure. In addition, it appears to have been developed for non-medical personnel. Are there studies where this has been validated in medical personnel?
  7. For the Copenhagen burnout inventory, more info on the reliability and validity should be provided
  8. More information on the reliability and validity of the turnover intention scale should be provided
  9. Line 120: 117 responses is lower than the power analysis said you needed; why didn’t you continue data collection until you reached 150?
  10. Line 136-137 typo
  11. Line 179-180 What type of analyses did you do to find these results?
  12. With all of these data, what about shift length and timing? That plays a huge role in burnout and emotional labor, but that isn’t even mentioned and varies widely with different types of nurses.
  13. Line 247 typo
  14. Table 3, 4 and 5 are too big and don’t make much sense. It is way too much to interpret. Why not give some of the info in the form of a figure?
  15. The discussion (section 4.1) is great, but it is all a restatement of the study’s findings and does not include any additional information.
  16. Line 302 “The results of this survey indicate that the participants had been exposed to moderate to high levels of emotional labor, burnout, and turnover intention.”  What does this mean?  They experienced it? Or were exposed to it?
  17. In the abstract, you say, “The results 24 suggest that adverse mental health outcomes and their consequences can be improved at the hospital level by coordinating work departments and improving work environments.”  This is in no way supported by the study and results.

Overall this can be revamped and has important information, but some major issues should be addressed.

Author Response

  • Response to Comments from Reviewer 2

Overall comment:

This paper describes a part of a larger project assessing factors related to burnout, emotional labor, medical errors and turnover intention in nurses in South Korea.  The paper addresses issues that are pertinent to healthcare, and it includes new tasks that have not been evaluated in the past (as well as a novel population).

Overall, the paper has some merit, but there are some major issues to be addressed. The biggest concern is lack of background and support for the measures, as well as the small sample size. In addition, one of the biggest factors related to burnout in healthcare is shift-work, and that is not mentioned at all.  For the participants, the type of certification is listed, but we know nothing about the shift types, duration, frequency, etc.  That is a major factor that should be considered.

Response:              

Thank you for your valuable comments on our manuscript.

  1. The background and support for the measures has been strengthened. (It is described in more detail below.)
  2. We added an explanation for the small sample size and efforts to overcome it and emphasized it as a major limitation in the Conclusions. (It is described in more detail below.)
  3. The analysis related to shift work was already included in this study, but it seems that we did not describe it sufficiently. We have added a shift-related analysis to this revised version. (It is described in more detail below.)

Comment 1:

Lit review is too short. They mention turnover intention and the difficult job nurses have, but they suddenly jump to medical errors and burnout without talking about history related to factors affecting those.  Definitions of emotional labor and burnout, as well as previous studies tying them together and detailing how they affect nurses, should be expanded.

Response:              

Thank you for your comment. In the Introduction section, we emphasized the importance of this topic and previous studies that could constitute a potential rationale for this study.

“Accordingly, the Korean Nurses Association held a nursing policy declaration ceremony in 2016 with the slogan ‘Happy Nurses Make Happy People. Securing skilled nurses for patient safety and preventing turnover was one of the main objectives. Thus, it is evident that the mental health of nurses and prevention of the resultant turnover intention are of much significance in Korea. … High burnout in nurses is associated with high turnover intention, and empirical research is needed to better understand the so-called sustainability at work and to prevent burnout and turnover in this population [10]. Van der Heijden et al. (2019) presented a model of the order of job demands (including emotional demands), burnout, and turnover intention, and emphasized the need to understand the turnover intention of nurses from the individual, organizational, and social perspectives [10]. However, few studies examine the relationships among emotional labor, burnout, turnover intention, and medical error among nurses, especially in the South Korean context. In particular, existing studies have often investigated the relationship between emotional labor, burnout, and turnover intention, but studies considering medical errors are lacking. Examinations of these factors can ultimately help relieve burdens at the individual, hospital, and societal levels by improving the individual mental health of nurses and nursing practice environments [10]. Specifically, a previous study suggested that reducing the quantitative job demands and increasing social support for nurses was associated with a decrease in turnover intention [10], suggesting the possibility of managing nurse turnover intention at the social rather than the individual level. These strategies should be developed to reflect the sociocultural context [11]. Moreover, the COVID-19 pandemic may have discriminatory effects depending on the country, suggesting that the mental health of nurses should be considered contextually [12].”

(please see pages 1 to 2, red words)

Comment 2:

Line 55“Examinations of these factors can ultimately help relieve burdens at the individual, hospital, and societal levels by improving the individual mental health of nurses and nursing practice environments.”  With this statement, you assume we are aware of burdens at individual, hospital and societal levels, but you have provided no information.

Response:              

Thank you for the comment. We have added information to support this statement.

“Examinations of these factors can ultimately help relieve burdens at the individual, hospital, and societal levels by improving the individual mental health of nurses and nursing practice environments [10]. Specifically, a previous study suggested that reducing the quantitative job demands and increasing social support for nurses was associated with a decrease in turnover intention [10], suggesting the possibility of managing nurse turnover intention at the social rather than the individual level. These strategies should be developed to reflect the sociocultural context [11]. Moreover, the COVID-19 pandemic may have discriminatory effects depending on the country, suggesting that the mental health of nurses should be considered contextually [12].”

(please see page 2, red words)

Comment 3:

Line 58“Our research team is participating in a project that looks to develop strategies by which to improve the mental health of South Korean nurses working in hospital settings.” This information needs to be touched on in the lit review. There is no background provided on this or what mental health issues are in South Korean nurses.

Response:              

Thank you for your comment. We have added literature to support our study objective and have provided a background clarifying why mental health issues are important and which issues are of particular significance to Korean nurses.

“The research team consists of the authors of this paper participating in a project that looks to develop strategies (i.e., smartphone application using mind-body medicine) to improve the mental health of South Korean nurses working in hospital settings. Recent findings of this project included the effectiveness of mind-body modalities on the mental health of nurses [13]. In this study, with reference to the model proposed by Van der Heijden et al. [10], the relationships between emotional labor, burnout, and turnover intention were investigated, and another important outcome, medical error, among nursing staff in a university hospital in South Korea, was also considered. This hospital is the only university hospital that provides both Western and Korean medicine services (i.e., integrative medicine), where staff are likely to be familiar with mind-body medicine in Busan, the second largest city in South Korea.”

(please see page 2, red words)

Comment 4:

Line 58 paragraph starts using first person, which is unusual in a primary research article

Response:              

Thank you for your comment. We apologize for the inappropriate English style. We have made amendments throughout the manuscript to avoid first-person expressions.

Comment 5:

Why doesn’t the participants section have the actual number of participants and demographics on them?

Response:              

Thank you for your comment. We had omitted this information for the purpose of condensing the content of the earlier manuscript, but we have added this data back in the revised version.

“A total of 117 responses were collected and analyzed after excluding five incomplete responses (completion rate: 95.9%; 117/122). Most participants were women (n = 113; 96.6%) and under 30 years of age (n = 65; 55.6%). Most of the participants had a bachelor degree or higher (n = 87; 74.4%), and more than two-thirds (n = 78; 66.7%) were unmarried. Most of the participants were nonreligious (n = 72; 61.5%).”

(please see page 5, red words)

Comment 6:

The emotional labor measure should have more information on the reliability and validity of the measure. In addition, it appears to have been developed for non-medical personnel. Are there studies where this has been validated in medical personnel?

Response:              

Thank you for your comment. We have cited a related clinical study to explain that this tool was validated for medical personnel, too.

“The emotional labor assessment tool of Lee was used to assess the study participants’ emotional labor [14]. This tool’s 14-item scale comprises two domains: employee-focused emotional labor (six items) and job-focused emotional labor (eight items). Each item is measured on a five-point Likert scale anchored by 1 (not at all) and 5 (very much); therefore, the score ranges of employee-focused emotional labor and job-focused emotional labor are 6–30 and 8–40, respectively. The higher the score was, the more severe the emotional labor. In Lee’s study on emotional labor of workers (not medical staff) [14], the values of Cronbach’s α for employee-focused emotional labor and job-focused emotional labor according to subscales were 0.78–0.80 and 0.79–0.92, respectively. In addition, the model suggested by Lee showed a goodness of fit to the empirical data (goodness of fit [GFI] = 0.90; root mean square residual [RMR] = 0.04; normed fit index [NFI] = 0.79; and parsimonious normed fit index = 0.64) [14]. In a study using this assessment tool in emergency room nurses in Korea [15], the values of Cronbach’s α for employee-focused emotional labor and job-focused emotional labor according to subscales were 0.69–0.81 and 0.74–0.84, respectively, which are satisfactorily high.”

(please see page 3, red words)

Comment 7:

For the Copenhagen burnout inventory, more info on the reliability and validity should be provided.

Response:              

Thank you for your comment. We have added information about the reliability and validity of the Copenhagen Burnout Inventory.

“Ham’s Korean-language version of the Copenhagen Burnout Inventory [16] was used to assess participant burnout [17]. The 19-item scale comprises three domains: personal burnout (seven items), work-related burnout (six items), and client-related burnout (six items). Each item is measured on a five-point Likert scale anchored by 1 (not at all) and 5 (very often); therefore, the score ranges of personal burnout, work-related burnout, and client-related burnout are 7–35, 6–30, and 6–30, respectively. The higher the score was, the more severe the burnout. The Copenhagen Burnout Inventory is a useful measure to evaluate burnout, and a recent study confirmed that Cronbach’s α for personal burnout, work-related burnout, and client-related burnout for hospital nurses were 0.91, 0.89, and 0.92, respectively [18]. In addition, this particular study found data supporting the construct validity of this tool by using confirmatory factor analysis and moderate to high correlation with other measures, including overall work environment, job satisfaction, and intent to leave, suggesting its convergent validity [18]. In Ham’s study on Korean nurses working in university-based hospitals [17], Cronbach’s α for burnout was 0.93.”

(please see page 3, red words)

Comment 8:

More information on the reliability and validity of the turnover intention scale should be provided

Response:              

Thank you for your comment. We have added information about the reliability and validity of the Turnover Intention Scale.

“The Korean Nurse Turnover Intention Scale (K-NTIS) of Yeun et al. was used to assess participant turnover intention [19]. The 10-item scale comprises three domains: job satisfaction (four items), job performance (three items), and interpersonal relationships (three items). Each item is measured on a five-point Likert scale anchored by 1 (not at all) and 5 (very much); therefore, the score ranges of job satisfaction, job performance, and interpersonal relationships are 4–20, 3–15, and 3–15, respectively. The higher the score was, the more severe the turnover intention. In Yeun et al.’s study on Korean nurses working in general hospitals [19], the values of Cronbach’s α for job satisfaction, job performance, and interpersonal relationships were 0.83, 0.78, and 0.80, respectively. In addition, this study used confirmatory factor analysis to identify data supporting the construct validity and average variance extracted to identify data supporting the discriminant validity of this tool [19]. The model suggested by Yeun et al. showed a goodness of fit to the empirical data of GFI = 0.95; RMR = 0.04; and NFI = 0.92 [19].”

(please see pages 3 to 4, red words)

Comment 9:

Line 120: 117 responses is lower than the power analysis said you needed; why didn’t you continue data collection until you reached 150?

Response:              

Thank you for your comment. The authors requested cooperation from the nursing headquarters of this hospital on the 5th day of the survey to encourage participation of nursing staff, but the participation rate did not increase significantly. Therefore, after in-depth discussion among the researchers, the survey was concluded on the 10th day because it was thought that ongoing pressure to participate in the study could aggravate nurses’ work stress. This suggests that perhaps a sample size should have been developed that reflected the predicted participation rate as well as the survey response rate. We acknowledge that this is a major limitation, and has been included in the revised manuscript.

First, this study was conducted at a university hospital in South Korea. This hospital is unique in that it is the only university hospital that provides integrative medicine services in Busan, the second largest city in South Korea. However, this also means that these participants do not represent the entire Korean nursing staff population. In addition, the sample size calculated before the survey was set to 150, in consideration of the potential poor response, but the final number of participants was 117, which is less than the required sample size of 134. The researchers requested cooperation from the nursing headquarters of this hospital on the 5th day of the survey to encourage participation of nursing staff, but the participation rate did not increase significantly. Therefore, through in-depth discussion among the researchers, the survey ended on the 10th day because it was thought that ongoing pressure to participate in the study could aggravate the nurses’ work stress. Therefore, it is acknowledged as a major limitation that this research did not reach the target sample size.”

(please see page 15, red words)

Comment 10:

Line 136-137 typo

Response:              

Thank you for the comment. This typo has been corrected.

Comment 11:

Line 179-180 What type of analyses did you do to find these results?

Response:              

Thank you for your comment. The levels of emotional labor, burnout, turnover intention, and medical errors were analyzed using descriptive statistics.

“The data collected in this study was analyzed using SAS® version 9.4 (SAS Institute. Inc., Cary, NC, USA). The demographic characteristics are expressed herein as either frequencies and percentages or means and standard deviations (SDs). The levels of emotional labor, burnout, turnover intention, and medical errors were analyzed using descriptive statistics.”

(please see page 4, red words)

Comment 12:

With all of these data, what about shift length and timing? That plays a huge role in burnout and emotional labor, but that isn’t even mentioned and varies widely with different types of nurses.

Response:              

Thank you for your comment. Naturally shift work was considered, but it seems that it was not described sufficiently in the original manuscript. This content has been added to in the revised manuscript.

“The demographic characteristics investigated in this survey included gender, age, clinical experience, education level, marriage, religion, monthly income, job position, work type, work department, whether they were working in a department they wished to work, whether they were satisfied with their current department, turnover experience, and department transfer experience; subjective health status on a five-point Likert scale anchored by very bad and very good, were also captured. It was also investigated whether working in specific shifts affected the results. The nurses here work in three shifts: day, evening, and night, starting at 7 am, 3 pm, and 11 pm, respectively.”

(please see page 3, red words)

“However, there were no statistically significant differences in all outcomes, including emotional labor, burnout, medical error, and turnover intention levels in terms of age, marital status, religion, and type of work (shift or non-shift) (all p >.05) (Table S2).”

(please see page 8, red words)

“Contrary to what might be expected, no significant differences were found in emotional labor, burnout, medical error, and turnover intention levels according to the type of work (i.e., shift work or non-shift work). However, Lee et al. [26] suggested that working night shifts is a risk factor for burnout in nurses working in general hospitals. Another study found that shift nurses experienced sickness presenteeism more frequently than non-shift nurses [27]. One possibility for this discrepancy is that shift work may be associated with poor mental health through the mediation of sleep disturbances that were not investigated in this study [28]. Other possibilities include the potential mediation of other contributors to shift work tolerance, including age, morningness, self-esteem, social support, job stress, alcohol consumption, physical activity, and number of night shifts [29].”

(please see page 14, red words)

Comment 13:

Line 247 typo

Response:              

Thank you for the comment. This typo has been corrected.

Comment 14:

Table 3, 4 and 5 are too big and don’t make much sense. It is way too much to interpret. Why not give some of the info in the form of a figure?

Response:              

Thank you for your suggestion. It was difficult to convert the results in tables 3-5 into a figure as they were the results of the regression analysis. Instead, we have shortened the existing tables to the greatest extent possible and presented the originals in the Supplementary Tables.

Comment 15:

The discussion (section 4.1) is great, but it is all a restatement of the study’s findings and does not include any additional information.

Response:              

Thank you for your comment. We had listed this information to summarize our study findings, but we have now removed it in response to the reviewers' comments that it was redundant.

Comment 16:

Line 302 “The results of this survey indicate that the participants had been exposed to moderate to high levels of emotional labor, burnout, and turnover intention.”  What does this mean?  They experienced it? Or were exposed to it?

Response:              

Thank you for your comment. We agree this sentence lacked clarify. It has been modified to read as:

“The results of this survey indicated that the participants had moderate to high levels of emotional labor, burnout, and turnover intention.”

(please see page 13, red words)

Comment 17:

In the abstract, you say, “The results 24 suggest that adverse mental health outcomes and their consequences can be improved at the hospital level by coordinating work departments and improving work environments.”  This is in no way supported by the study and results.

Response:              

Thank you for your comment. We agree with this comment. We corrected the sentence expressions that were not supported by the research results as follows:

“These results can be used to improve the mental health outcomes, and their consequences, of nurses working in the hospital. Specifically, the job positions of nursing personnel may be a major consideration in such a strategy, and job-focused emotional labor and employee-focused emotional labor may be promising targets in ameliorating turnover intention and client-related burnout, respectively.”

(please see page 1, red words)

Comment 18:

Overall, this can be revamped and has important information, but some major issues should be addressed.

Response:              

Thank you for your comment. I think your comments have made our research more valuable, and as guide to future research in the area.

Round 2

Reviewer 1 Report

Congratulations! The manuscript has been improved and your research is valuable for publication. The results are better presented now, the literature references and debates are now engaged with recent debates in the theoretical literature . Good luck wth your future work and research in the area!

Author Response

  • Response to Comments from Reviewer 1

Overall comment:

Congratulations! The manuscript has been improved and your research is valuable for publication. The results are better presented now, the literature references and debates are now engaged with recent debates in the theoretical literature . Good luck wth your future work and research in the area!  

Response:              

Thank you very much for your warm comments.

Reviewer 2 Report

Thank you for your extensive revisions and your explanations of the changes. I appreciate how much you added, and I feel that the paper, as well as the support for this research, is much stronger.  I have two comments/suggestions for changes.

  1. I appreciate that you added validity and reliability information for the measures (as I asked for), but you don't necessarily have to add as MUCH info as you did. It is fine like that for each of the measures, but even just saying that X study in X year confirmed the validity is fine (you just didn't have ANYTHING the first time.
  2. One concern I had was the type of statistical analysis you did for the items that are now on lines 232-256.  In the data analysis portion you say you use one way ANOVAs and independent t-tests, but I don't know which ones you used here, and if you did ANOVAs, where are the post-hoc analyses? I feel like more description is needed on this paragraph, because it is unclear exactly what test you did.  I think this is a fairly simple fix, as in saying, "A one-way analysis of variance (ANOVA) determined X", but you have multiple analyses you are running, so it is unclear how many tests you ran, whether your probability of error is increased due to doing multiple independent t tests, etc.  This is my only major concern left with this paper.

I feel that if you address the statistical analysis question that this paper would be a great addition to the literature.  Perhaps you have a biostatistician you can consult?

Author Response

  • Response to Comments from Reviewer 2

Overall comment:

Thank you for your extensive revisions and your explanations of the changes. I appreciate how much you added, and I feel that the paper, as well as the support for this research, is much stronger.  I have two comments/suggestions for changes.

Response:              

Thank you for your valuable comments on our manuscript.

Comment 1:

I appreciate that you added validity and reliability information for the measures (as I asked for), but you don't necessarily have to add as MUCH info as you did. It is fine like that for each of the measures, but even just saying that X study in X year confirmed the validity is fine (you just didn't have ANYTHING the first time.

Response:              

Thank you for your comment. We shortened the sentence according to your comment.

“2.2.2. Emotional labor

The emotional labor assessment tool of Lee was used to assess the study participants’ emotional labor [14]. This tool’s 14-item scale comprises two domains: employee-focused emotional labor (six items) and job-focused emotional labor (eight items). Each item is measured on a five-point Likert scale anchored by 1 (not at all) and 5 (very much); therefore, the score ranges of employee-focused emotional labor and job-focused emotional labor are 6–30 and 8–40, respectively. The higher the score was, the more severe the emotional labor. The validity of this tool was confirmed in Lee’s study in 2007 [14]. Also, its reliability was confirmed by Lee (not medical staff) [14] and Eo et al. (emergency room nurses) [15].

2.2.3. Burnout

Ham’s Korean-language version of the Copenhagen Burnout Inventory [16] was used to assess participant burnout [17]. The 19-item scale comprises three domains: personal burnout (seven items), work-related burnout (six items), and client-related burnout (six items). Each item is measured on a five-point Likert scale anchored by 1 (not at all) and 5 (very often); therefore, the score ranges of personal burnout, work-related burnout, and client-related burnout are 7–35, 6–30, and 6–30, respectively. The higher the score was, the more severe the burnout. The Copenhagen Burnout Inventory is a useful measure to evaluate burnout, and Ham [17] and Montgomery et al. [18] confirmed its validity and reliability in nurses.

2.2.4. Turnover intention

The Korean Nurse Turnover Intention Scale (K-NTIS) of Yeun et al. was used to assess participant turnover intention [19]. The 10-item scale comprises three domains: job satisfaction (four items), job performance (three items), and interpersonal relationships (three items). Each item is measured on a five-point Likert scale anchored by 1 (not at all) and 5 (very much); therefore, the score ranges of job satisfaction, job performance, and interpersonal relationships are 4–20, 3–15, and 3–15, respectively. The higher the score was, the more severe the turnover intention. The validity and reliability of this tool was confirmed by Yeun et al. in Korean nurses working in general hospitals [19].”

(please see page 3, red words)

Comment 2:

One concern I had was the type of statistical analysis you did for the items that are now on lines 232-256.  In the data analysis portion you say you use one way ANOVAs and independent t-tests, but I don't know which ones you used here, and if you did ANOVAs, where are the post-hoc analyses? I feel like more description is needed on this paragraph, because it is unclear exactly what test you did.  I think this is a fairly simple fix, as in saying, "A one-way analysis of variance (ANOVA) determined X", but you have multiple analyses you are running, so it is unclear how many tests you ran, whether your probability of error is increased due to doing multiple independent t tests, etc.  This is my only major concern left with this paper.

Response:              

Thank you for the comment.

This research is a study to investigate the correlation between emotional labor, burnout, medical error, and turnover intention and logistic regression analysis was performed.

Table S2 shows the process of finding variables to be used as control variables prior to logistic regression analysis. If there were 2 items of each variable, independent t-test was used, if there were 3 or more items, ANOVA was used, and Bonferroni's post hoc test was performed. If there was a significant difference between the items, it was used as a control variable for logistic regression analysis, which includes clinical experience, education level, job position, working department, the department where they desired to work, the department where they satisfy to work, turnover experience, department transfer experience, and subjective health (Table 3-4, Table S4-6).

In summary, we did not present the results of post-hoc analysis because we did not evaluate how much difference there is between the items of the variable, but used it as a control variable if there is a significant difference.

These statistical issues are based on the knowledge of one of the authors, a statistician (O.-J.K.). In this revised version, we have briefly supplemented the above description. If the reviewer think that this is not enough, we would like to mention that we are always happy to receive your further comments. Thank you.

“The differences in emotional labor, burnout, turnover intention, and medical error according to demographic characteristics were further analyzed using independent t-tests and analysis of variance (ANOVA). To examine correlations among emotional labor, burnout, turnover intention, and medical error, Pearson’s correlation analysis was performed. The variable used for correlation analysis was the average of the scores of the Likert type questionnaire, and after confirming normality through the Shapiro-Wilk Kolmogorov-Smirnov tests, Pearson's correlation coefficient was obtained. In addition, it was confirmed that the analysis results using the Spearman and Pearson correlation coefficients were not significantly different in this study. Finally, multiple regression analysis was performed to determine the factors that affect participant turnover intention and medical error, and multicollinearity in the multiple regression analysis was checked for. Control variables were identified prior to logistic regression analysis, by using independent t-test or ANOVA, and Bonferroni's post-hoc test was performed. The significance level of the statistical test was set at p < .05.”

(please see page 4, red words)

Comment 3:

I feel that if you address the statistical analysis question that this paper would be a great addition to the literature.  Perhaps you have a biostatistician you can consult?

Response:              

One of the authors, a statistician, would like to respond to your comment as above. However, we always welcome your further comments. Thank you.
